# The ameliorative effect of pioglitazone against colistin-induced nephrotoxicity is mediated by inhibition of NF-κB and restoration of Nrf2 signaling: An integrative bioinformatics prediction-guided in vitro study

**Metab Alharbi**[1]ⓘ◉*, **Mohamed A. Mahmoud**[1]ⓘ◉, **Abdulrahman Alshammari**[1], **Mashal M. Almutairi**[1], **Jihan M. Al-Ghamdi**[2], **Jawza F. Alsabhan**[3], **Othman Al Shabanah**[1], **Norah A. Alshalawi**[1], **Sami I. Alzarea**ⓘ[4], **Abdullah F. Alasmari**[1]

1 Department of Pharmacology and Toxicology, College of Pharmacy, King Saud University, Riyadh, Saudi Arabia, 2 Biochemistry Department, College of Science, King Saud University, Riyadh, Saudi Arabia, 3 Department of Clinical Pharmacy, College of pharmacy, King Saud University, Riyadh, Saudi Arabia, 4 Department of Pharmacology, College of Pharmacy, Jouf University, Sakaka, Aljouf, Saudi Arabia

◉ These authors contributed equally to this work.

* mesalharbi@ksu.edu.sa

## Abstract

Pioglitazone, an anti-diabetic drug, has been previously shown to ameliorate kidney damage through anti-inflammatory and antioxidant effects. In this study, we employed an integrative bioinformatics approach to study the possible mechanisms involved in the mitigative effect of pioglitazone against colistin-induced nephrotoxicity. Next, we validated the results obtained from the bioinformatics study by pre-treating human kidney-2 (HK-2) cell line with pioglitazone 100 $\mu$M for 30 minutes and then treating the cells with colistin sulfate 1200 $\mu$M for 24 hours. Inflammatory signaling by cytokines and the nuclear factor erythroid 2 related factor 2 (Nrf2) signaling pathways were predicted to be involved in the ameliorative effect of pioglitazone against colistin-induced nephrotoxicity. The nuclear factor kappa B subunit p65 (NF-κB p65) and Nrf2 were among the predicted transcription factors regulating the hub genes. Moreover, miR-24, miR-16, and miR-21 were identified as potential pathogenic miRNAs regulating the hub genes. In contrast, miR-17, miR-27a, and miR-146a were identified as potential protective miRNAs. The *in vitro* study indicated that pioglitazone pre-treatment increased cell viability in HK-2 cells exposed to colistin. Pioglitazone pre-treatment reduced the expression of pro-inflammatory cytokine genes (IL6 and TNF). Moreover, pioglitazone reduced the protein expression of NF-κB p65 and increased the protein expression of Nrf2. The protective effect of pioglitazone against colistin-induced toxicity in HK-2 cells is related to its anti-inflammatory and antioxidant activity through modulating NF-κB-mediated inflammatory signaling and Nrf2-mediated antioxidative stress signaling.

**Data Availability Statement:** All relevant data are within the manuscript and its Supporting Information files.

**Funding:** The authors extend their appreciation to the Deputyship for Research & Innovation, "Ministry of Education" in Saudi Arabia for funding this research work through the project number (IFK-SUDR_H162)". The funder has provided the needed fund to buy the materials and equipment for this work to be done.

**Competing interests:** The authors have declared that no competing interests exist.

# Introduction

The past two decades have witnessed colistin's revival as a last resort treatment against multidrug resistant gram-negative bacteria. However, the use of colistin is associated with the development of dose-limiting nephrotoxicity that can precipitate treatment failure [1]. The significance of colistin-induced nephrotoxicity is evident in a recently published meta-analysis of five randomized controlled trials including more than 300 patients [2]. In this study, colistin-induced nephrotoxicity occurred at 36.2%, and the risk of nephrotoxicity development was 140% compared to β-lactam antibiotics. The suboptimal dosing of colistin is widely reported and often leads to the emergence of bacterial resistance, treatment failure, and mortality [3]. Only half of the patients treated with colistin achieve a steady-state plasma concentration sufficient to kill bacteria due to adverse renal effects, despite having normal renal function before therapy. Hence, it is important to study the feasibility of administering an agent alongside colistin to attenuate or abolish its nephrotoxicity without reducing its anti-bacterial activity. Consequently, this intervention has the potential to widen the therapeutic window and allow for the administration of optimal doses of colistin.

The accumulation of colistin in kidney tubular cells that highly express the transporters megalin, human peptide transporter 2, and carnitine/organic cation 2 is a prerequisite for colistin-induced kidney injury [1]. Once colistin gains access to the intracellular compartment of renal proximal tubular epithelial cells (RPTECs), it invokes mitochondrial dysfunction probably through interactions with cardiolipin, a lipid found in mitochondrial membranes [4]. Colistin induces nephrotoxicity by activating multiple pathologic pathways, including the unfolded protein response, mitochondrial apoptotic pathways, and nuclear events leading to oxidative stress, DNA damage, inflammation, and apoptosis [5]. A recently undertaken metabolomic and transcriptomic study carried out in rats treated with colistin methanesulfonate, a prodrug of colistin, at a high dose (50 mg/kg) has revealed that colistin kidney toxicity leads to an imbalance between pro-oxidant and antioxidant elements, resulting in oxidative stress and pro-inflammatory signaling [6]. In another study, a gene networking analysis has revealed that the levels of nuclear factor erythroid-2-related factor-2 (Nrf2)- mediated oxidative stress genes and glutathione-mediated detoxification signaling were downregulated in colistin-treated rats as opposed to untreated control animals [7]. In a different study, Shafik and colleagues reported that colistin treatment of rats at a dose of 300,000 IU/Kg for six days upregulated the expression of PH domain and leucine-rich repeat protein phosphatase leading to the reduced expression of p-Akt and activation of GSK3β [8]. Active GSK3β upregulates Fyn kinase expression that phosphorylates Nrf2 leading to Nrf2 nuclear export to the cytosol and consequent Nrf2 ubiquitination and degradation in the cytosol. Reduced Nrf2 expression leads to diminished reduced glutathione (GSH) content and blunted superoxide dismutase (SOD) activity and increased caspase-3 leading to renal toxicity. In another study, Shafik et al. reported that colistin treatment of rats at a dose of 300,000 IU/Kg per day for six days reduced the expression of Nrf2 compared to the control animals [9]. The authors linked the reduced Nrf2 expression to the downregulation of miR-205 expression leading to an increased expression of Egl-9 family hypoxia-inducible factor 2 (EGLN2). The increased expression of EGLN2 increases the degradation of hypoxia-inducible factor-1 leading to reduced Nrf2 expression and blunting the GSH content and reducing SOD activity. Dai et al. have implicated oxidative stress in playing a critical role in colistin-induced nephrotoxicity. The study in mice reported that colistin treatment at a dose of 18 mg/kg/day for seven days reduced the level of the crucial antioxidant enzymes SOD and catalase (CAT) [10]. Moreover, the malondialdehyde, a by-product of lipid peroxidation, was increased. Additionally, the authors reported decreased GSH level. In the same study, the authors reported an increased expression of nuclear factor-κB (NF-κB) [10].

Furthermore, the levels of pro-inflammatory cytokines interleukin-1 beta (IL-1β) and tumor necrosis factor alpha (TNF-α) were also increased. The authors implicated increased reactive oxygen species (ROS) and downregulation of the antioxidant status in the upregulation of NF-κB and the consequent infiltration of inflammatory cells.

Pioglitazone, an anti-diabetic drug, works by activating the nuclear receptor peroxisome proliferator-activated receptor gamma (PPAR-γ). Apart from its role as an insulin sensitizer, pioglitazone possesses mitochondrial protective and regenerative effects [11]. Pioglitazone improves mitochondrial respiration, reduces mitochondrial ROS, and increases adenosine tri-phosphate production, and oxygen consumption rate [12]. Studies have shown that pioglitazone can shore up cellular antioxidant defenses through direct binding to the PPAR-γ receptor [13]. Nrf2 and PPAR-γ crosstalk forms a positive feedback loop through which each transcriptional factor reinforces the expression of the other and increases the expression of target genes encoding antioxidant enzymes such as CAT, SOD, Glutathione peroxidase 3, and heme oxygenase-1 (HO-1). Additionally, PPAR-γ activation is linked to the induction of an anti-inflammatory response by the transrepression of the pro-inflammatory transcriptional factor NF-κB, preventing it from mediating the expression of inflammatory mediators TNF-α, IL-1β, and interleukin-6 (IL-6) [13]. Further, pioglitazone has been shown to activate the silent information regulator 1 (sirt1), a histone deacetylase ubiquitously present in the cytosol and nucleus of RPTECs [14, 15]. Upregulation of sirt1 reduces the acetylation and hence the activity of several transcriptional factors that contribute to colistin-induced nephrotoxicity, including p53, NF-κB, and forkhead box protein O.

Based on the abovementioned activities, it is hypothesized that pioglitazone may exert a protective effect against colistin-induced nephrotoxicity. Therefore, using an integrative bioinformatics prediction approach we aimed to investigate the potential mechanisms underpinning the protective effect of pioglitazone treatment in the context of colistin-induced nephrotoxicity. Thereafter, we aimed to confirm the protective effect by studying the effect of pioglitazone pre-treatment on colistin-induced cytotoxicity in human kidney-2 (HK-2) cells, a well-established *in vitro* model for RPTECs cytotoxicity, and investigate markers related to the pathways predicted by the bioinformatics study.

## Materials and methods

### Gene data sources

The potential targets of AKI were retrieved from DisGeNET (https://www.disgenet.org) [16] by inputting "acute kidney injury" into the search box. The potentials targets of pioglitazone were collected from CTD (https://ctdbase.org/) [17] by choosing "chemicals" and inserting "pioglitazone" into the search box. The online tool Draw Venn Diagram (http://bioinformatics.psb.ugent.be/webtools/Venn/) was used to identify potential therapeutic targets for PIO against AKI. The functional classification of therapeutic targets was performed by Panther database (accessed in September 2024) [18].

### STRING enrichment and functional module analysis

The intersection genes between AKI and pioglitazone were treated as potential genes involved in the ameliorative effect of pioglitazone against AKI. The official genes symbols of the intersection genes were inserted into the STRING website (https://string-db.org/) [19] under the "Multiple Proteins" option and the organisms was set to "*Homo Sapiens*". Next, the attained PPI network was exported to Cytoscape software (version 3.10.1 Boston, MA, USA) [20] and the topological parameters of the network was retrieved using the "Analyze Network" option. The subnetwork was identified using the MCODE algorithm with the degree cut-off = 2, node

score cut-off = 0.2, k-core = 2, and maximum depth = 100. The top 20 genes in the attained subnetwork were identified using CytoHubba according to the MNC, Degree, EPC, Closeness, and Radiality node ranking options. The attained CytoHubba genes of the subnetwork were considered as the hub genes for pioglitazone and AKI interaction.

## Gene ontology and pathway enrichment analysis

The official hub genes symbols were inserted into DAVID (https://david.ncifcrf.gov/) [21, 22] and the species was set as "*Homo Sapiens*". Next, the "Functional Annotation Tool" was selected and "Gene Ontology" and "Reactome" options were checked. The BPs, CCs, and MFs were attained and the top 10 terms in each category were plotted in an enrichment bubble. Significant terms ($p < 0.05$) were included in each category. Next, the top 30 significant reactome pathways ($p < 0.05$) were plotted in an enrichment bubble. The enrichment bubble plots were plotted using the SRplot tool (https://www.bioinformatics.com.cn/srplot) [23].

## Upstream regulatory network analysis

The official hub genes symbols were inserted into the X2K web (https://maayanlab.cloud/X2K/) [24, 25] database and the TFs, PKs, and the intermediate proteins, which potentially regulate the hub genes were obtained. The attained X2K network was exported to Cytoscape and the MCODE algorithm was used to identify the subnetworks with the parameters set at degree cut-off = 2, node score cut-off = 0.2, k-core = 2, and maximum depth = 100.

## MicroRNA regulatory network analysis

The official hub genes symbols were inserted into the NetworkAnalyst (https://www.networkanalyst.ca/) [26, 27] database under the gene list input option. The organism type was set to "*Homo Sapiens*" and the ID type was set to "Official Gene Symbols". Next, the gene-miRNA interaction option was chosen and the miRTarBase version 8 database [28] was applied. Only miRNAs with at least 5 connections to the hub genes were included. The gene-miRNA interaction network was exported to Cytoscape to visualize the miRNA network that potentially regulates the hub genes.

## Reagents and kits

Colistin sulfate, pioglitazone hydrochloride, Dulbecco's modified eagle's medium (DMEM), fetal bovine serum (FBS), penicillin-streptomycin 10,000 U/mL, dimethyl sulfoxide (DMSO), phosphate buffer saline (PBS), MTT were purchased from Sigma Chemical Company (St. Louis, MO, USA). Radioimmunoprecipitation assay buffer (RIPA), Halt™ protease and phosphatase inhibitor, Pierce™ bicinchoninic acid (BCA) protein assay kit, TRIzol Reagent, High-capacity cDNA reverse transcription kit, SYBR green master mix were purchased from Thermo Fisher Scientific (Waltham, MA, USA). 10x Tris/Glycine/SDS, 10x Tris/glycine, 10x Tris Buffered Saline (TBS), Clarity™ Western ECL Substrate, and 4x Laemmli Sample Buffer were purchased from Bio Rad (Hercules, CA, USA). IL-6, TNF-α, and glyceraldehyde-3-phosphate dehydrogenase (GAPDH) primers were purchased from Integrated DNA Technologies (Coralville, IA, USA). NF-κB p65, Beta actin, and GAPDH antibodies were purchased from ELK Biotechnology (Denver, CO, USA). Nrf2 antibody was purchased from Santa Cruz Biotechnology (Dallas, TX, USA). HRP-conjugated Goat Anti-Rabbit lgG and HRP-conjugated Goat Anti-Mouse lgG were purchased from ELK Biotechnology (Denver, CO, USA).

## Cell culture and treatments

Immortalized HK-2 cells (CRL-2190 ™) were acquired from American Type Culture Collection (Manassas, VA, USA). The cells were cultured in DMEM media supplemented by 10% FBS and 1% penicillin-streptomycin. The cells were maintained in a cell culture incubator at 37°C under a humidified 5% $CO_2$ atmosphere, and the medium was renewed every three days. The cells were treated as follows: The control group was treated with serum free medium. The pioglitazone-treated group was treated with pioglitazone 100 μM for 30 minutes and then treated with serum free media for 24 hours. The colistin-treated group was treated with serum free media for 30 minutes and then colistin sulfate 1200 μM for 24 hours. The combination group was treated with pioglitazone 100 μM for 30 minutes and then colistin sulfate 1200 μM for 24 hours.

## Cell viability assay

HK-2 cells were seeded in 96-well microplates at a density of $3 \times 10^3$ cells per well. After cell treatment, 15 $\mu$L 0.5% (w/v) MTT reagent in serum free media was added to each well in the dark, and plates were incubated for four hours in the cell culture incubator. The entire medium was removed, and the formazan crystals were solubilized by the addition of 200 $\mu$L acidified isopropyl alcohol to each well, and the absorbance was read at a wavelength of 570 nm by a microplate reader (BioTek Epoch 2, Agilent, Santa Clara, CA, USA). The results were reported as % cell viability normalized to the control group.

## RNA extraction and RT-qPCR

The mRNA expression of IL-1β, IL-6, TNF-α, NQO1, and HO-1 was assessed using RT-qPCR. GAPDH was used as an internal control. HK-2 cells were seeded in 6-well plates at a density of $1.2 \times 10^5$ cells per well and treated as mentioned before. The total cellular RNA was extracted from HK-2 cells in each treatment group using the TRIzol reagent as previously described [29]. The total RNA concentration and purity (260/280 ratio) were measured by NanoDrop 8000 Spectrophotometer (Thermo Fisher Scientific, Waltham, MA, USA). 2 $\mu$g total RNA was used to synthesize cDNA using the High-capacity cDNA reverse transcription kit according to the manufacturer's manual. The SYBR green master mix was used to quantify the mRNA expression of IL-1β, IL-6, TNF-α, NQO1, HO-1, and GAPDH. The specific gene primer sequences are shown in (**Table 1**). The RT-qPCR was done using Applied Biosystems 7500 Real-Time PCR (Thermo Fisher Scientific, Waltham, MA, USA) and the cycles were 95°C for 10 minutes, 40 cycles of 95°C for 15 seconds, 60°C for 30 seconds, and 72°C for 30 seconds, 1 cycle (melt curve) 95°C for 15 seconds, 60°C for 60 seconds, and 95°C for 15 seconds. The obtained comparative threshold cycle values for each group was normalized by the comparative threshold cycle value of GAPDH and the fold change was calculated by the $2^{-\Delta\Delta Ct}$ method as previously described [30]. The results were reported as a ratio of control.

**Table 1. Gene specific primer sequences.**

| Gene | Orientation | Primer sequence (5' to 3') |
|------|-------------|---------------------------|
| IL6 | Forward | AGACAGCCACTCACCTCTTCAG |
| | Reverse | TTCTGCCAGTGCCTCTTTGCTG |
| TNF | Forward | CTCTTCTGCCTGCTGCACTTTG |
| | Reverse | ATGGGCTACAGGCTTGTCACTC |
| GAPDH | Forward | GTCTCCTCTGACTTCAACAGCG |
| | Reverse | ACCACCCTGTTGCTGTAGCCAA |

## Protein extraction and western blotting

The protein expression of NF-κB p65 and Nrf2 was assessed by western blotting. Beta actin and GAPDH were used as a loading control. HK-2 cells were seeded in 6-well plates at a density of $1.2 \times 10^5$ cells per well and treated as mentioned before. The cells were lysed by RIPA buffer containing 1% protease and phosphatase inhibitor for 30 minutes and then centrifuged at 14,000g at 4˚C for 20 minutes. The protein concentration of the supernatant was determined by BCA method as previously described [31]. Equal protein concentration was mixed with 4x Laemmli sample buffer containing 5% 2-mercaptoethanol and boiled for 10 minutes in a water bath. 15 $\mu$g protein from each sample was loaded to sodium dodecyl-sulphate polyacrylamide gel and electrophoretically resolved on 10% gels in 1x Tris/Glycine/SDS buffer at 80 volts for 30 minutes and then 120 volts for 1.5 hours. The separated protein was electrophoretically transferred in 1x Tris/glycine buffer containing 20% methanol to methanol pre-activated polyvinylidene difluoride membranes using the semi-dry method at 25 volts for 45 minutes. The membranes were blocked by 3% fat free milk in 1x TBS with 0.1% ($v/v$) tween-20 for one hour at room temperature. The membranes were washed three times with 1x TBS with 0.1% ($v/v$) tween-20 (washing buffer) for five minutes each and then incubated with primary antibodies against NF-κB p65 (1: 1000), Nrf2 (1: 2000), Beta actin (1:1000), and GAPDH (1: 1000) overnight at 4˚C on a shaker. The membranes were washed three times with washing buffer for five minutes each and incubated with horseradish peroxidase conjugated secondary antibody (1: 5000) in 3% fat free milk solution in washing buffer for one hour at room temperature. The blots were washed three times with washing buffer for five minutes each and incubated with western ECL Substrate for one minute and then imaged with ChemiDoc (Bio Rad, Hercules, CA, USA). The density of the blots was measured by image J software and normalized by GAPDH and the fold change in protein expression was calculated relative to the control. The results were reported as ratio of control.

## Statistical analysis

The results were expressed as means ± standard error of mean. The normality of the data was tested by the Shapiro–Wilk test and *p-value* > 0.05 was considered as normal distribution. Homogeneity of variance was tested using Brown-Forsythe test with *p-value* > 0.05 considered homogenous. The normally distributed data with homogenous variance was analyzed by ordinary one-way analysis of variance followed by the Tukey-Kramer test for multiple comparisons using the GraphPad software version 10.1.1 (La Jolla, CA, USA). Results were considered statistically significant if the *p-value* was < 0.05.

## Results

### Retrieval and intersection of AKI-associated genes and PIO-related genes and classification of intersection genes

185 acute kidney injury (AKI)-related genes (S1 Table in S1 File) were identified using Disease-Gene Network (DisGeNET), a database containing disease and gene association information. Next, 821 pioglitazone-related genes (S2 Table in S1 File) were retrieved from the Comparative Toxicogenomic Database (CTD). 44 intersection genes associated with AKI and pioglitazone were identified (**Fig 1A**) and classified into 12 categories (**Fig 1B**), including RNA metabolism protein (NOS2 and NOS3), cell adhesion molecule (CAM1), chromatin/chromatin-binding (RB1), cytoskeletal protein (CRP), gene-specific transcriptional regulator (HIF1A, NFE2L2, PPARG, and AR), intracellular signal molecule (IL1B, EDN1, SPP1, TNF, IGF1, INS, VEGFA, and TGFB1), metabolite interconversion enzyme (NOS3, NQO1, HMOX1, NOS2,

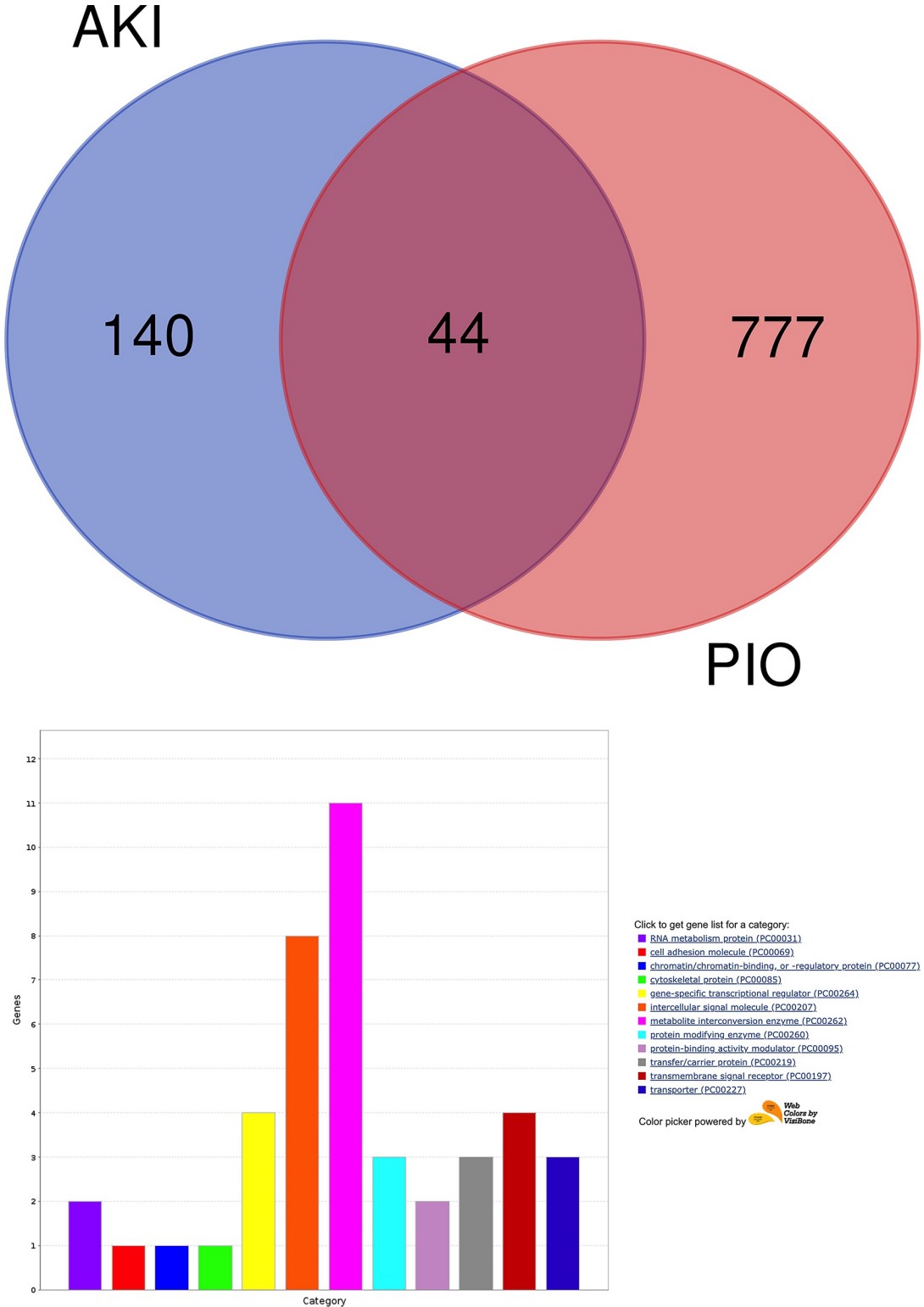

**Fig 1.** **(a)** Venn diagram showing the intersection genes between AKI-associated genes and PIO-related genes. **(b)** Functional classification of the intersection genes according to the Panther database.

AR, SOD1, GSTP1, CYP2E1, UGT1A1, SPP1, and PTGS2), protein modifying enzyme (GSK3B, ACE, and MTOR), protein-binding activity modulator (CDKN1B and CCND1), transfer/carrier protein (GC, TF, and MB), transmembrane signal receptor (TF, EGF, MET, and AVPR1A), and transporter (SLC22A2, CACNA1S, and ATP5F1B).

## STRING enrichment analysis and functional module analysis

The intersection genes were used to perform Search Tool for the Retrieval of Interacting Genes (STRING) enrichment analysis (**Fig 2A**). The attained protein-protein interaction (PPI) network was analyzed by Cytoscape to determine the network topological parameters and uncover the subnetwork (**Fig 2B**). The STRING network contained 41 nodes, 402 edges, and the average number of neighbours was 20.1. The subnetwork was unmasked by the Molecular Complex Detection (MCODE) plugin and only one subnetwork containing 25 nodes and 268 edges was attained. The CytoHubba plugin identified the top 20 genes in the uncovered sub-network using the Maximum Neighborhood Component (MNC), Degree, Edge Percolated Component (EPC), Closeness, and Radiality options. The top 20 genes identified included (TNF, IL6, HIF1A, IL1B, BCL2, INS, PTGS2, PPARG, IL10, HMOX1, TGFB1, SPP1, NOS3, MTOR, IGF1, ICAM1, GSK3B, EGF, NFE2L2, and CRP). The CytoHubba genes were considered as hub genes for subsequent bioinformatics analysis.

## Gene ontology and pathway enrichment analysis

The official hub genes symbols were inserted to the Database for Annotation, Visualization and Integrated Discovery (DAVID) and the biological process (BP), cellular compartment (CC), molecular function (MF) tables were obtained. Only significant terms ($p < 0.05$) were considered for each category (252 BPs, 13 CCs, and 18 MFs) (S3, S4, and S5 Tables in S1 File). The top 10 enriched terms in each category are shown in the enrichment bubble plot (**Fig 3A**). Remarkably, inflammatory response and positive/negative regulation of apoptosis are highly enriched BPs. Whereas extracellular space and extracellular region are highly enriched CCs. The highly enriched MFs included cytokine activity. A reactome pathway enrichment analysis table was also obtained from DAVID. The significantly ($p < 0.05$) enriched pathways are shown in (S6 Table in S1 File). The top 30 enriched reactome pathways are shown in the enrichment bubble plot (**Fig 3B**). Notably, regulation of HMOX1 expression and activity, NFE2L2 regulating antioxidant/detoxification enzymes, nuclear events mediated by NFE2L2, and KEAP-NFE2L2 pathway were highly enriched reactome pathways. In addition, interleukin -4, -13, and -10 signaling, signaling by interleukins, cytokine signaling in immune system were highly enriched reactome pathways.

## Upstream regulatory network analysis

The eXpression2Kinases (X2K) Web database was used to identify the transcription factors (TFs) and protein kinases (PKs) regulating the expression of the hub genes. The top 10 TFs (S7 Table in S1 File) and the top PKs (S1 Fig in S1 File) are shown in supplementary material. In addition, the upstream regulatory network of the hub genes is shown (**Fig 4A**). Notably, RELA and NFE2L2 are among the top 10 TFs regulating the hub genes. Using the MCODE plugin in Cytoscape three clusters were obtained from the upstream regulatory network. In cluster 1 (**Fig 4B**), MAPK1, also known as ERK2, appears to be an important PK regulating RELA. In cluster 2 (**Fig 4C**), MAPK14, also known as p38α, is the only PK in the network connected through intermediate proteins to four different TFs (SMC3, CBX3, TRIM28, and E2F1). In cluster 3 (**Fig 4D**), three PKs (CSNKK2A1, DNAPK, and CDK2) and three TFs (CTCF, RAD21, and NFE2L2) were identified.

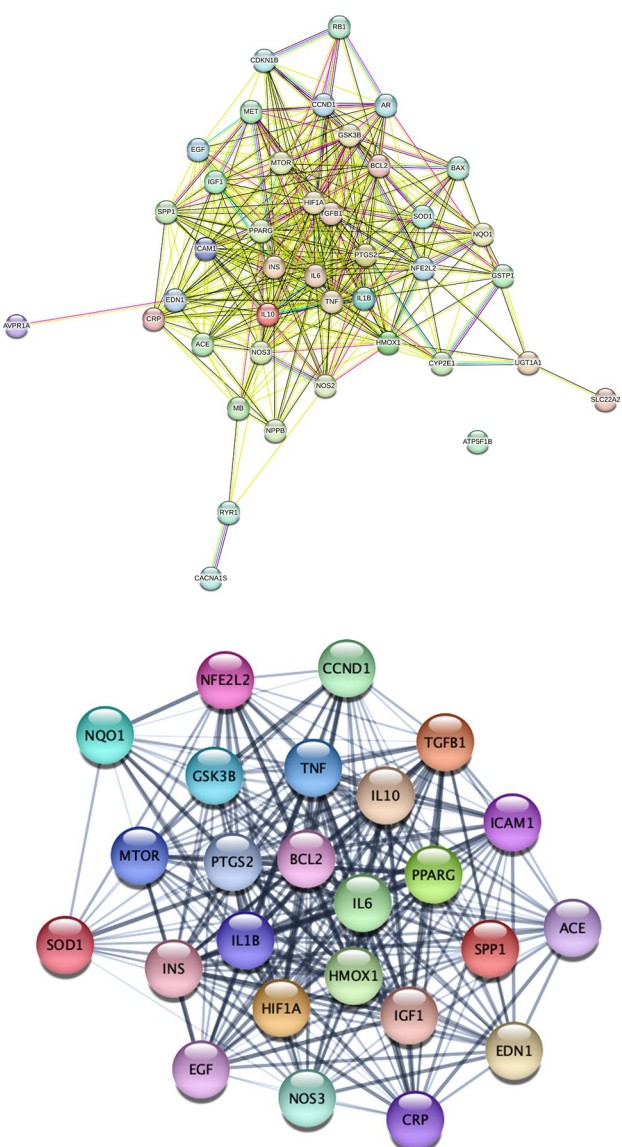

**Fig 2.** (**a**) STRING PPI network of AKI and pioglitazone intersection targets; (**b**) The subnetwork uncovered by Cytoscape containing 25 nodes and 268 edges.

## MicroRNA regulatory network analysis

The NetworkAnalyst database was used to uncover the MicroRNAs (miRNAs) regulating the hub genes. The top 10 miRNAs based on the number of connections to hub genes are shown (S8 Table in S1 File). The top 2 miRNAs (miR-24-3p and miR-335-5p) had six connections to hub genes. The rest of identified miRNAs (miR-16-5p, miR-17-5p, miR-21-5p, miR-26b-5p, miR-27a-3p, miR-98-5p, miR-106a-5p, and miR-146a-5p) had five connections. Next, the NetworkAnalyst generated network was exported to Cytoscape to display the network (**Fig 5**).

## Effect of pioglitazone pre-treatment on colistin-treated HK-2 cells viability

To confirm the results attained from the bioinformatics study, we investigated the effect of pioglitazone pre-treatment on HK-2 treated cells. Firstly, we studied the effect of colistin

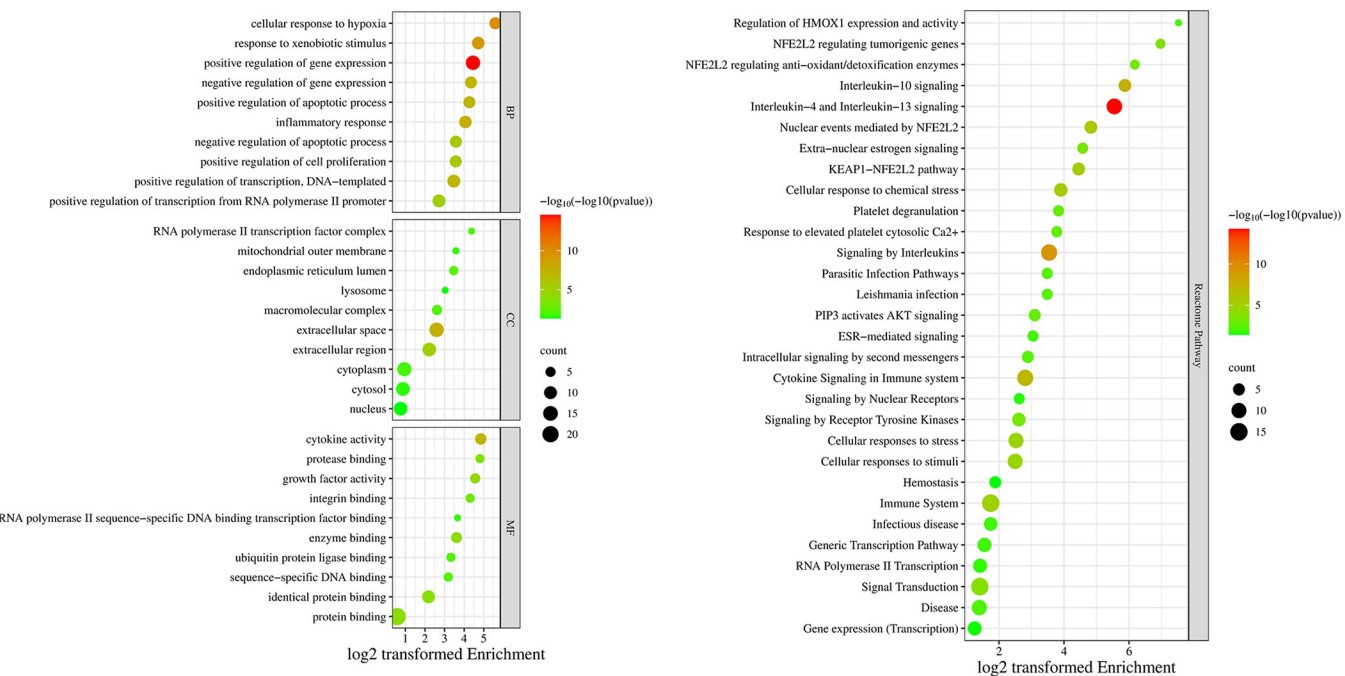

**Fig 3. (a)** Enrichment bubble plot of top 10 enriched BPs, CCs, and MFs; **(b)** Top 30 reactome pathways plotted in enrichment bubble.

treatment on HK-2 cells viability by treating the cells with different concentrations of colistin (500 $\mu$M, 750 $\mu$M, 1000 $\mu$M, 1500 $\mu$M, and 2000 $\mu$M). The concentrations of colistin were determined from a previous study, which reported that the half-maximal inhibitory concentration ($IC_{50}$) of colistin against RPTECs is in the high micromolar range (250–690 $\mu$M) [32]. In our study, colistin treatment significantly reduced cell viability in a concentration-dependent manner with an $IC_{50}$ close to 750 $\mu$M (**Fig 6A**). Next, we studied the effect of pioglitazone pre-treatment on colistin-treated HK-2 cells. Pioglitazone pre-treatment time was determined from a previous study [15], which reported that pre-treatment of HK-2 cells with pioglitazone 100 $\mu$M for 30 minutes significantly increased cell viability in HK-2 cells exposed to cisplatin. In our study, we pre-treated HK-2 cells for 30 minutes with different concentrations of pioglitazone (100 $\mu$M, 150 $\mu$M, and 200 $\mu$M) and then treated the cells with colistin 1200 $\mu$M (the colistin concentration was chosen to reduce cell viability to 20%). All three pioglitazone concentrations significantly increased cell viability compared to colistin-only treated cells (**Fig 6B**). However, the higher pioglitazone concentrations (150 $\mu$M and 200 $\mu$M) did not show a significant increase in cell viability compared to 100 $\mu$M; therefore, we used 100 $\mu$M in subsequent experiments.

## Effect of pioglitazone pre-treatment on colistin-induced upregulation of IL-6 and TNF-α gene expression

To ascertain whether pioglitazone affects the expression of pro-inflammatory cytokines as predicted from the bioinformatics study. We studied the effect of pioglitazone *100 $\mu$M* pre-treatment for 30 minutes on the mRNA expression of IL6 and TNF in HK-2 cells treated with colistin 1200 $\mu$M for 24 hours. Using quantitative real time polymerase chain reaction (RT-qPCR) we confirmed that pioglitazone pre-treatment significantly reduced IL6 (*p < 0.05*) (**Fig 7A**) and TNF (*p < 0.001*) (**Fig 7B**) mRNA levels compared to colistin-treated HK-2 cells.

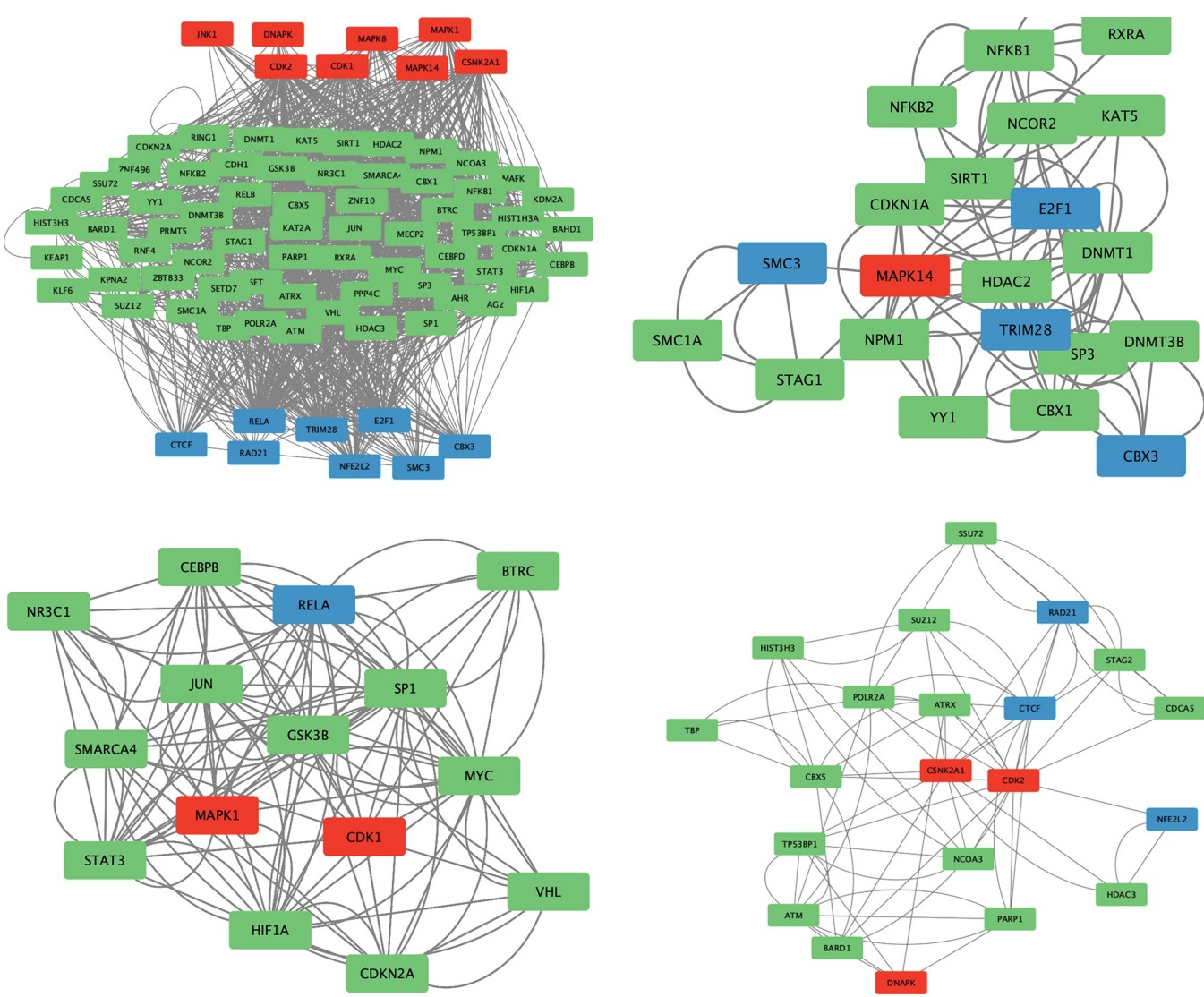

**Fig 4.** **(a)** Upstream regulatory network; **(b)** Cluster 1; **(c)** Cluster 2; **(d)** Cluster 3. TFs (blue), intermediate protein (green), PKs (red).

### Effect of pioglitazone pre-treatment on colistin-induced increase in NF-κB p65 and decrease in Nrf2 protein expression

The protein expression of NF-κB p65 and Nrf2 was analyzed by western blotting in cells pre-treated with pioglitazone 100 $\mu M$ or serum free media for 30 minutes and then exposed to colistin 1200 $\mu M$ for 24 hours. Pioglitazone pre-treatment significantly reduced ($p < 0.05$) (**Fig 8A**) the expression of NFκB p65 and increased ($p < 0.05$) (**Fig 8B**) the expression of Nrf2 compared to colistin-only treated cells.

## Discussion

Colistin has emerged as a drug of last resort in the post-antibiotic era. Lack of experience with dosing, narrow therapeutic index, and risk of mortality among intensive care patients contribute to the challenges faced by physicians when confronted with colistin-induced nephrotoxicity [33]. In addition, this nephrotoxicity is associated with a high economic burden due to an

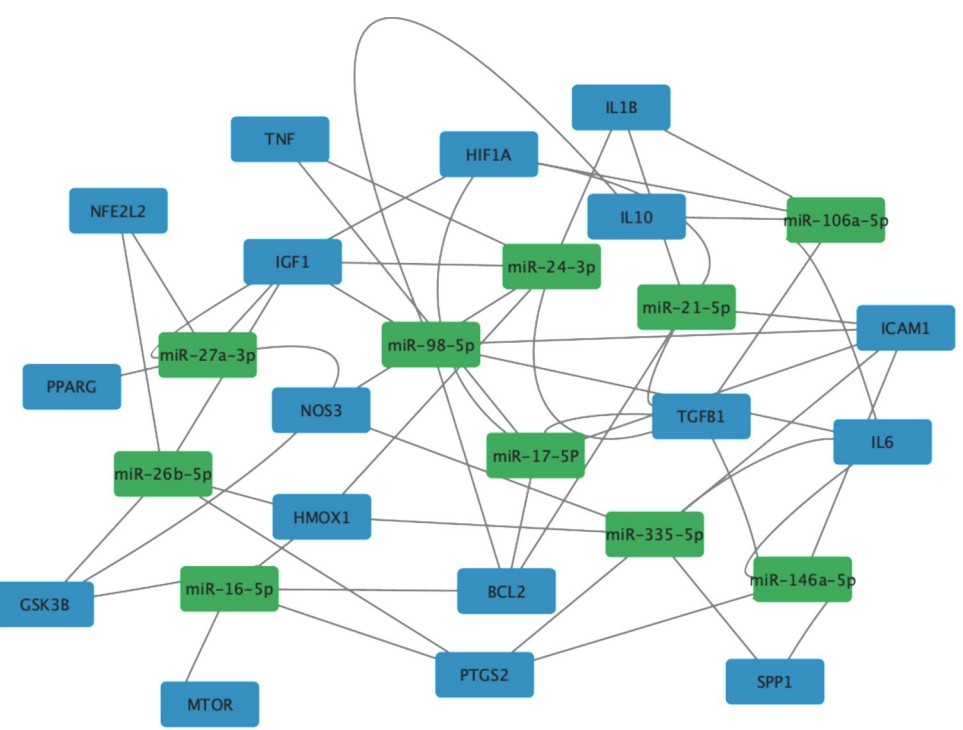

**Fig 5. miRNA regulatory network of hub genes.** Genes (blue), miRNA (green).

increased hospital length of stay and hemodialysis [34]. Therefore, it is vital to investigate the potential mitigative effect of pioglitazone against this limiting adverse effect. The results obtained from the integrative bioinformatics study identified two main signaling pathways regulated by the hub genes, including inflammatory cytokines signaling and the Keap1-Nrf2 pathway. Among the TFs in the upstream regulatory network of the hub genes were RELA and NFE2L2 genes, which encode the TFs NF-κB p65 and Nrf2, respectively. Moreover, as predicted by the upstream regulatory network of the hub genes, pioglitazone pre-treatment reduced protein expression of NF-κB p65 and increased the protein expression of Nrf2, indicating that pioglitazone exerts anti-inflammatory and antioxidative stress activity against colistin-induced toxicity in HK-2 cells. Furthermore, in agreement with the bioinformatics data, pioglitazone reduced pro-inflammatory cytokines (IL-6 and TNF-α) gene expression in colistin-treated HK-2 cells. These effects were associated with increased cell viability in pioglitazone pre-treated HK-2 cells exposed to colistin.

The bioinformatics analysis of the hub genes revealed that cytokine signaling plays a major role in the interaction between pioglitazone and AKI. All the three aspects of gene function reported by GO enrichment analysis indicated that cytokines are critical in AKI and pioglitazone interaction. Cytokine activity was the most enriched MF of the intersection genes. The extracellular space and region were highly enriched CCs; the sites where secreted pro-inflammatory cytokines bind to surface receptors to induce downstream signaling. The inflammatory response was one of the highly enriched BPs. Moreover, the reactome pathway confirmed the importance of cytokine signaling, particularly IL-4, IL-13, and IL-10 signaling. IL-4 and IL-13 signaling are essential for recovery from AKI by modulating macrophage polarization from a pro-inflammatory M1 phenotype to the reparative M2 phenotype [35], suggesting that pioglitazone anti-inflammatory activity may stem from its ability to modulate macrophage activity. Moreover, IL-10 signaling is protective in different models of AKI, including cisplatin-induced

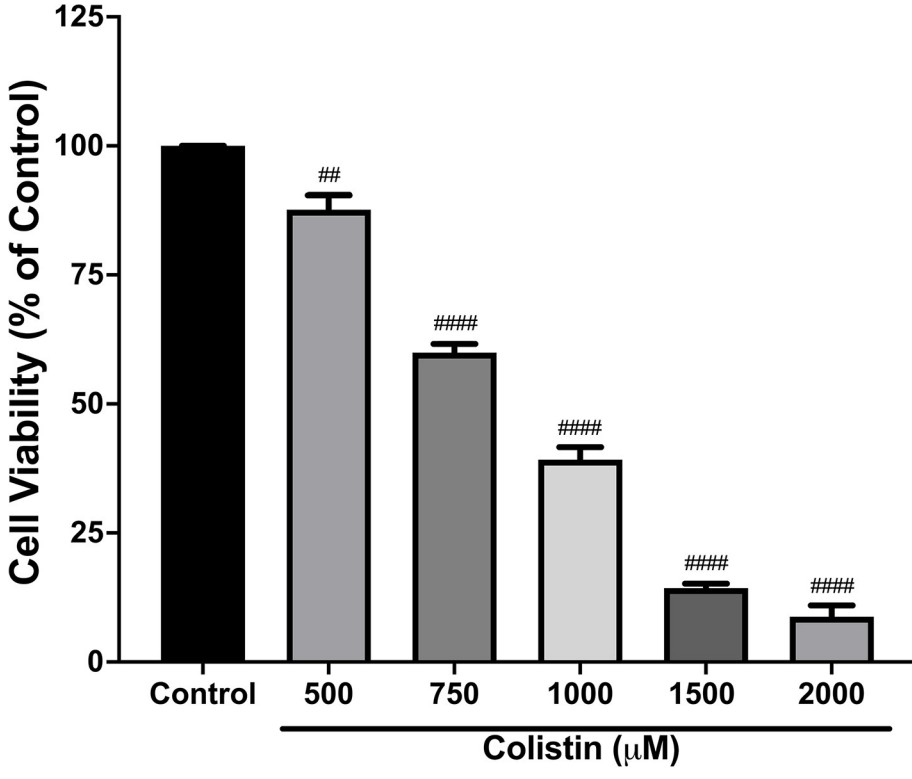

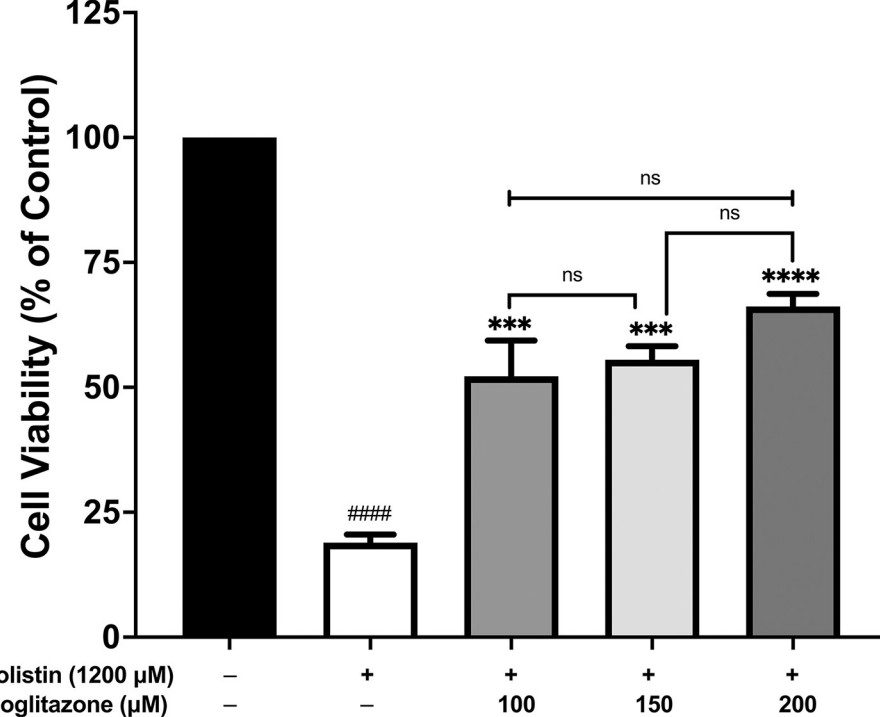

**Fig 6. Pioglitazone increased cell viability in HK-2 cells treated with colistin compared to colistin-only treated cells.** The cell viability was assessed with 3-(4,5-dimethylthiazol-2-yl)-2,5-diphenyl-tetrazolium bromide (MTT) assay (a) HK-2 cells were treated with different concentrations of colistin (500 $\mu$M- 2000 $\mu$M) for 24 hours; **(b)**. HK-2 cells were treated with colistin 1200 $\mu$M alone or were pre-treated with different concentration of pioglitazone (100 $\mu$M, 150 $\mu$M, and 200 $\mu$M) for 30 minutes and then treated with colistin 1200 $\mu$M for 24 hours. Data mean ± SEM (n = 3).

$^{\#\#}p < 0.01$, $^{\#\#\#\#}p < 0.0001$ vs. the control group, $^{***}p < 0.001$, $^{****}p < 0.0001$ vs. the colistin-treated group, $^{ns}p > 0.05$. Statistical analysis was performed with one-way ANOVA with Tukey's multiple comparisons tests to compare the significance between treatment groups.

nephrotoxicity, ischemia-reperfusion injury (IRI), and endotoxin-induced AKI [36]. IL-10 possesses both anti-inflammatory and immunomodulatory activities, by increasing macrophage polarization to the reparative M2 phenotype, thereby reducing M1 macrophages-mediated release of pro-inflammatory cytokines (IL-1β and IL-6) [36, 37]. Pioglitazone treatment increases macrophages polarization to the anti-inflammatory M2 phenotype in mice kidney through its effect on PPAR-γ and activation of miR-23 that acts to repress interferon regulatory factor 1 and PBX/Knotted 1 Homeobox 1, which are crucial for macrophage M1 polarization [38]. Pioglitazone reduced the mRNA levels of pro-inflammatory cytokines (IL-6 and TNF-α) and increased the mRNA levels of anti-inflammatory cytokines (IL-4 and IL-10) in an *in vitro* model of calcium oxalate monohydrate-induced nephrotoxicity [38].

Another major pathway associated with the hub genes is the Kelch-like ECH-associated protein 1 (Keap1)-Nrf2 pathway. Three different activities related to Keap1-Nrf2 pathway, including regulation of HMOX1 expression and activity, NFE2L2 regulating antioxidant/detoxification enzymes, and nuclear events mediated by NFE2L2 were highly enriched reactome pathways in the intersection gene set. The Keap1-Nrf2 pathway is activated in stress conditions, particularly in response to ROS accumulation [39]. Under normal conditions, Nrf2 is bound to Keap1 in the cytosol, which targets it to proteasomal degradation; however, under oxidative stress conditions, this interaction is weakened and Nrf2 stability increases, allowing its nuclear translocation to mediate the expression of antioxidant genes (NQO1 and HMOX1) [39]. The central role of Nrf2 signaling in cisplatin-induced AKI has been shown in a recent study undertaken in HK-2 cells and in a mice model of cisplatin nephrotoxicity [40]. Cisplatin treatment of both HK-2 cells and mice reduced Nrf2 protein expression and its downstream antioxidant enzymes (NQO1 and HMOX1) compared to control groups, and this was associated with increased renal toxicity [40]. However, treatment with ammonium tetrathiomolybdate reduced the proteasomal degradation of Nrf2, leading to an increased Nrf2-mediated expression of antioxidant enzymes (NQO1 and HMOX1) and alleviating cisplatin nephrotoxicity in both models [40]. Previous studies have shown that pioglitazone treatment can activate the Keap1-Nrf2 pathway in rat brain and hepatic tissue [41, 42]. The exact mechanism of pioglitazone activation of Keap1-Nrf2 pathway remains to be elucidated.

The upstream regulatory network of the hub genes included the TFs NF-κB subunit p65 and Nrf2. NF-κB subunit p65 is a component of the NF-κB heterodimer involved in the activation of inflammation-related genes, including IL1B, IL6, and TNF [43]. Under normal conditions, NF-κB is sequestered in the cytosol by the interaction with the inhibitor of kappa B alpha (IκBα). Upon phosphorylation by inhibitor of kappa B kinases (IκBKs), IκBα is ubiquitinated and targeted for degradation, allowing the NF-κB heterodimer to translocate to the nucleus to positively regulate pro-inflammatory cytokines expression [44]. Our analysis revealed that ERK2, a mitogen activated protein kinase (MAPK), is upstream of NF-κB subunit p65 in cluster 1. Previous studies have shown that inhibition of ERK activation is associated with decreased NF-κB p65 signaling and reduced pro-inflammatory cytokines levels in different models of AKI [45–51]. In another study, Gong et al. have demonstrated that ERK phosphorylation coincides with reduced IκBα level and increased NF-κB p65 nuclear translocation [52]. Treatment with an ERK inhibitor reduced NF-κB p65 nuclear translocation, indicating that ERK activation is required for NF-κB p65 activation [52]. In cluster 2, p38α is upstream of CDKN1A (p21), a cyclin dependent kinase inhibitor involved in cell cycle arrest,

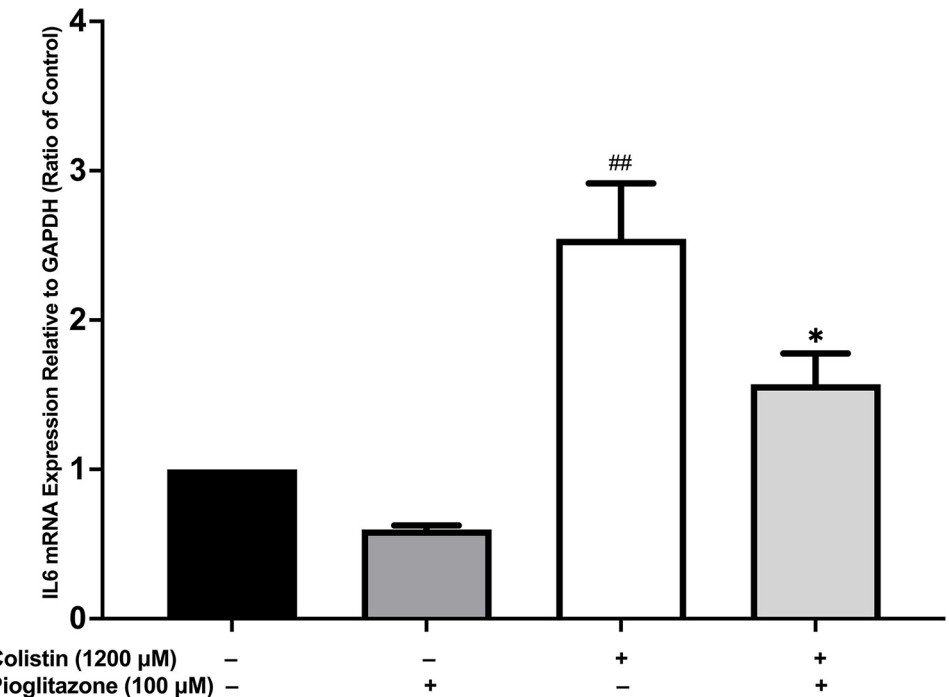

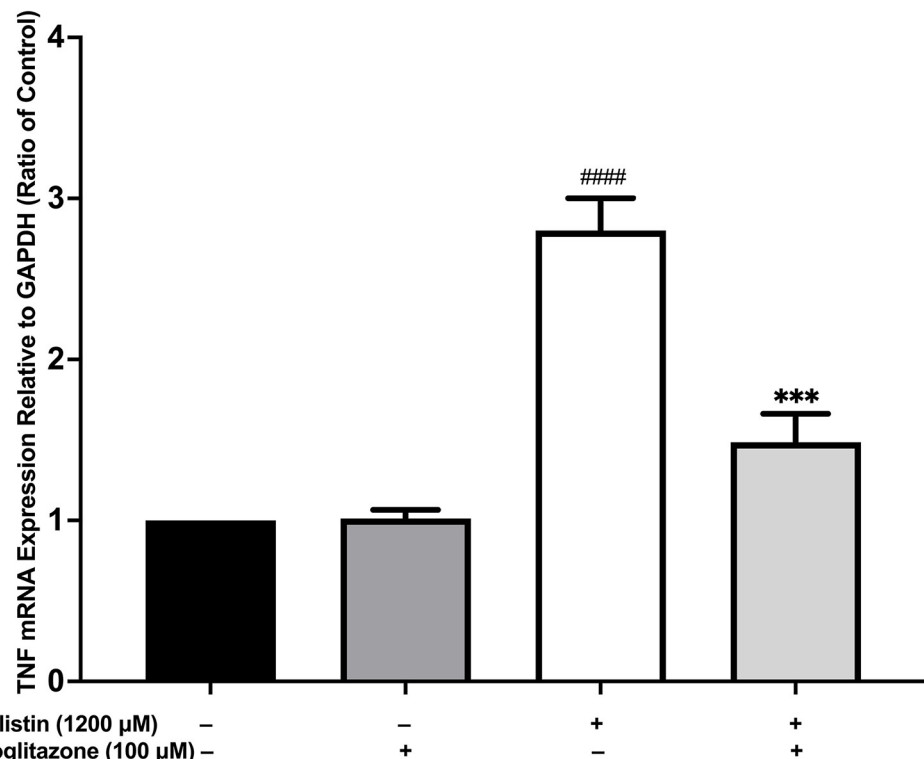

**Fig 7. Pioglitazone pre-treatment reduced the mRNA expression of IL6 and TNF in colistin-treated HK-2 cells compared to colistin-only treated cells.** The effect of pioglitazone 100 $\mu$M pre-treatment for 30 minutes on HK-2 cells treated with colistin 1200 $\mu$M for 24 hours on the mRNA expression of **(a)** IL6 and **(b)** TNF. Data mean ± SEM (n = 3). $^{##}p < 0.01$, $^{####}p < 0.0001$ vs. the control group, $^{*}p < 0.05$, $^{***}p < 0.001$ vs. the colistin-treated group. Statistical analysis was performed with one-way ANOVA with Tukey's multiple comparisons tests to compare the significance between treatment groups.

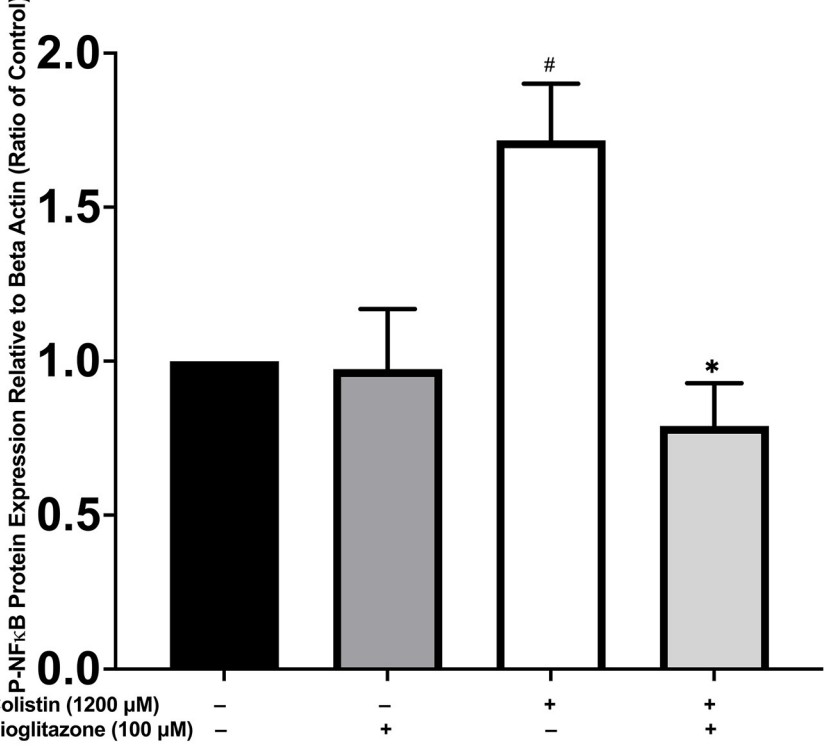

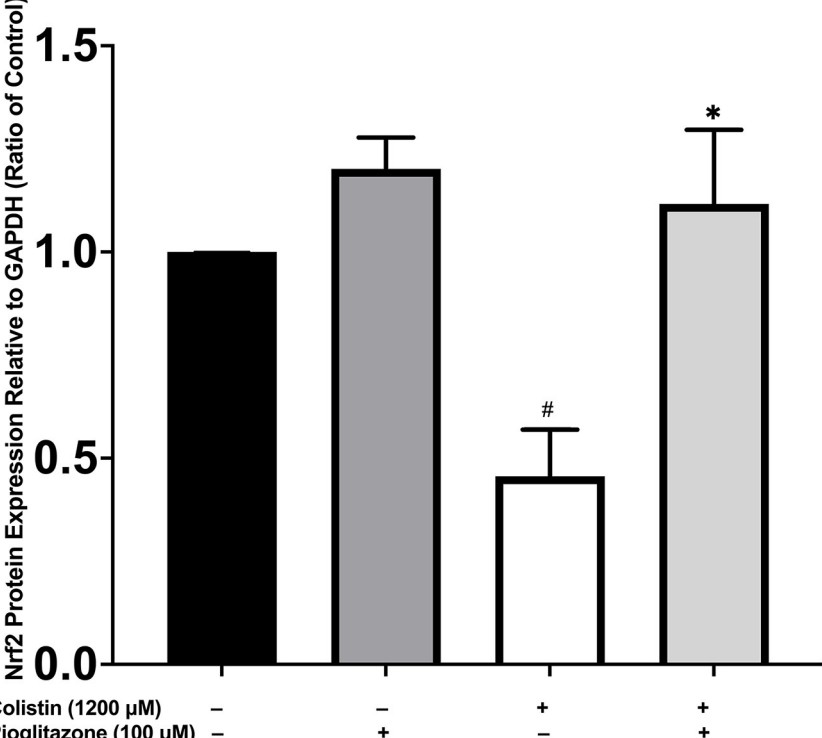

**Fig 8. Pioglitazone pre-treatment reduced the protein expression of NFκB p65 and increased the protein expression of Nrf2 in colistin-treated HK-2 cells compared to colistin-only treated cells.** The effect of pioglitazone 100 $\mu$M pre-treatment for 30 minutes on HK-2 cells treated with colistin 1200 $\mu$M for 24 hours on the protein expression of **(a)** NFκB p65 and **(b)** Nrf2. Data mean ± SEM (n = 3). [#]$p < 0.05$ vs. the control group, [*]$p < 0.05$ vs. the colistin-treated group. Statistical analysis was performed with one-way ANOVA with Tukey's multiple comparisons tests to compare the significance between treatment groups.

and E2F1, a TF that regulates the expression of genes involved in cell cycle progression. P38α is involved in cell fate decisions in stress conditions such as oxidative stress [53]. It is involved in cell cycle arrest in the initial phase of stress to reduce the replication of damaged DNA in renal cells [54]. P38α inhibits G1/S transition through stabilizing p21 and repressing the transcriptional activity of E2F1 [53]. In addition, p38α causes $G_2/M$ arrest by promoting the cytoplasmic retention of CDC25B/C phosphatases, which are responsible for removing phosphate residues on CDK1, leading to reduced CDK1 interaction with cyclin A and B and inhibiting mitotic entry and cell cycle progression through the mitotic phase [53]. In the final cluster, Nrf2 is associated with genes (ATM, ATRX, DNAPK, and RAD21) that encode DNA repair proteins, highlighting the role of Nrf2 in DNA repair and maintenance of genomic integrity under conditions of oxidative stress [55].

MiRNAs are non-protein coding RNAs that bind to the three prime untranslated region (3' UTR) of mRNA to post-transcriptionally regulate gene expression [56]. Recent evidence has highlighted the role of miRNAs in AKI and their potential value as therapeutic targets [56]. The regulatory effect of miRNAs on gene expression can contribute to the progression or amelioration of AKI [56]. The potential pathogenic miRNAs regulating the hub genes in our bioinformatics analysis included miR-24, miR-16, and miR-21. Lorenzen et al. have reported the association between renal IRI and the accumulation of miR-24, which directly reduces HMOX1 expression leading to increased apoptosis and greater renal injury [57]. In a different study, activation of CCAAT enhancer binding protein beta in renal IRI increased the expression of miR-16, which directly binds the 3' UTR of B-cell lymphoma (BCL2) attenuating BCL2 transcript translation leading to increased renal apoptosis and reduced renal function [58]. Huang and coworkers have demonstrated the pathogenic role of miR-21 in IRI-induced nephrotoxicity in HK-2 cells by reducing the activation of PI3K/AKT/mTOR pathway leading to increased pro-inflammatory cytokines release, elevated ROS, and apoptosis [59]. Pioglitazone treatment reduced miR-21 expression in another study, suggesting that pioglitazone may ameliorate AKI through targeting the pathogenic effect of miR-21 [60]. Our miRNA regulatory network analysis revealed potential protective miRNAs potentially regulating the hub genes, including miR-17, miR-27a, and miR-146a. All three protective miRNAs identified targeted NF-κB activation, albeit through different mechanisms. miR-17 negatively regulated the activation of NF-κB in lipopolysaccharide-treated HK-2 cells leading to reduced pro-inflammatory cytokines release, decreased apoptosis, and increased HK-2 cell viability [61]. miR-27a targeted inflammatory response in renal IRI through suppression of toll-like receptor 4 expression, which is upstream of NF-κB, thereby rescuing renal cell viability [62]. Exosomal miR-146a inhibited interleukin-1 receptor-associated kinase expression, reducing the phosphorylation of IκBα leading to reduced NF-κB p65 nuclear translocation and protecting HK-2 cells against hypoxia-reoxygenation injury [63]. The effect of pioglitazone treatment on expression of the identified potential pathogenic and protective miRNAs and their targets should be investigated in future studies.

We validated the findings obtained from the bioinformatics study in an *in vitro* model of colistin-induced nephrotoxicity. Our *in vitro* study in HK-2 cells confirmed the importance of NF-κB-mediated cytokine signaling in our model. Markó, Lajos, et al. have highlighted the pathogenic role of NF-κB activation in the tubular epithelium of mice subjected to IRI-induced kidney damage [64]. Genetic inhibition of NF-κB in the tubular epithelium attenuated renal IRI, reduced macrophage infiltration, and decreased apoptosis [64]. Dai and coworkers have reported that colistin treatment upregulated renal NF-κB expression and increased the levels of pro-inflammatory cytokines (IL-1β and TNF-α) in a mouse model of colistin-induced nephrotoxicity [10]. This is consistent with our findings in HK-2 cells which showed that colistin-treatment significantly increased the protein expression of NF-κB p65 and this coincided

with a significantly increased mRNA levels of pro-inflammatory cytokines (IL-6 and TNF-α), and reduced cell viability. Zhang et al. have reported that pioglitazone treatment prevented the activation of renal NF-κB p65 in mice treated with cisplatin through the adenosine monophosphate kinase-mediated deacetylation of NF-κB p65 [14]. In addition, the levels of pro-inflammatory cytokines (IL-1β, IL-6, and TNF-α) were reduced following pioglitazone treatment compared to cisplatin-only treated animals [14]. Similarly, our findings indicated that pioglitazone significantly reduced the expression of NF-κB p65 and significantly reduced the mRNA expression of pro-inflammatory cytokines (IL-6 and TNF-α) leading to increased cell viability. In a previous study, rosiglitazone, a PPAR-γ agonist similar to pioglitazone, reduced the phosphorylation and activation of NF-κB p65 in HK-2 cells exposed to cisplatin through a PPAR-γ-dependent mechanism [65]. Moreover, rosiglitazone protected mice from cisplatin-induced renal injury by reversing cisplatin-mediated depression of the anti-inflammatory cytokine IL-10 [66]. Rosiglitazone pre-treatment reduced renal macrophage infiltration, decreased caspase activation, and protected renal cells from apoptosis in this model [66].

The predicted involvement of Keap1-Nrf2 pathway in the protective effect of pioglitazone against colistin-induced nephrotoxicity was also validated in our *in vitro* study in HK-2 cells. Our data indicated that colistin treatment of HK-2 cells reduced the protein expression of Nrf2. This is consistent with a study undertaken by Wang et al. which indicated that colistin reduced the protein and gene expression of Nrf2 and HMOX1 and increased the expression of Keap1 in mouse renal tubular epithelial cells [67, 68]. Shafik and colleagues reported that colistin-treated rats at a dose of 300,000 IU/Kg for 6 days had a significantly reduced protein content of Nrf2 compared to control animals [8, 9]. In another study, Xiao and coworkers implicated polymyxin B (an antibiotic belonging to the same class as colistin)-mediated inhibition of Nrf2/NQO1 pathway in the induction of renal toxicity in *in vitro* and *in vivo* mouse models [69]. Pioglitazone pre-treatment of HK-2 cells exposed to colistin significantly increased Nrf2 protein expression. The effect of pioglitazone on the Nrf2 pathway in settings of AKI has not been reported in previous studies. However, pioglitazone activates the Nrf2 antioxidative stress pathway in several studies undertaken in *in vitro* and *in vivo* models of neuronal injury, supporting our finding in HK-2 cells [41, 70].

The mechanistic interplay between NF-κB and Nrf2 has been recently highlighted by Gao and colleagues [71]. The NF-κB and Nrf2 signaling pathways antagonize each other through complex mechanisms. For example, the NF-κB p65 inhibits Nrf2 transcriptional activity by hijacking CREB binding protein and recruiting histone deacetylase 3 causing histone hypoacetylation and reduced recruitment of Nrf2 to antioxidant response element [72]. HO-1, an antioxidant enzyme downstream of Nrf2, inhibits NF-κB signaling by increasing the levels of carbon monoxide and bilirubin (end products of heme catabolism) [71]. Prior studies have shown that several anti-inflammatory drugs that inhibit NF-κB activation can activate Nrf2 signaling, which might be the case with pioglitazone in our study [71]. Several lines of evidence have supported the major role played by the interplay between NF-κB and Nrf2 response in drug-induced nephrotoxicity. Qin et al. have reported that embelin treatment increased Nrf2 and reduced NF-κB signaling in cisplatin-induced nephrotoxicity [73]. In another study, treatment with the Nrf2 activator tertiary butylhydroquinone reduced NF-κB signaling in a diabetes aggravated model of renal IRI-induced AKI [74]. Vincamine ameliorated methotrexate-induced AKI through reducing NF-κB activation and increasing Nrf2 and HO-1 expression [75].

Our study is the first to report the potential protective effect of pioglitazone in HK-2 cells exposed to colistin sulfate. However, the results of this study should be treated as preliminary and further studies are required in *in vivo* models of colistin-induced nephrotoxicity to confirm the protective effect of pioglitazone. In addition, our bioinformatics analysis was based on

gene sets retrieved from databases and future studies should rely on RNA sequencing to identify the differentially expressed genes. Further, we did not validate the potential miRNAs regulating the hub genes and future studies should validate the potential pathogenic and protective miRNAs identified in this study and investigate the effect of pioglitazone on miRNAs expression and targets. The anti-inflammatory effect of pioglitazone is well established in literature; however, further studies are required to delineate the mechanisms of pioglitazone Nrf2 activation to test whether pioglitazone activates Nrf2 secondary to NF-κB inhibition or through a direct effect on Nrf2.

## Conclusion

To conclude, pioglitazone exerts nephroprotective activity against colistin through inhibition of NF-κB-mediated inflammatory response and activation of Nrf2-mediated antioxidative stress signaling. Our study is the first to report the protective effect of pioglitazone against colistin-induced nephrotoxicity. Our integrative bioinformatics tools predicted the involvement of inflammatory signaling by cytokines and the Nrf2 signaling pathways in the protective effect of pioglitazone against AKI. Our data indicates that colistin nephrotoxicity is associated with an imbalance in NF-κB and Nrf2 signaling leading to reduced cell viability. Pre-treatment with pioglitazone reverses the imbalance between NF-κB and Nrf2 signaling and mitigates colistin-mediated loss in cell viability.

## Supporting information

**S1 File.**
(DOCX)

**S1 Raw images.**
(PDF)

## Author Contributions

**Conceptualization:** Metab Alharbi.

**Data curation:** Metab Alharbi, Mohamed A. Mahmoud, Abdulrahman Alshammari, Mashal M. Almutairi, Jihan M. Al-Ghamdi, Jawza F. Alsabhan, Othman Al Shabanah, Norah A. Alshalawi, Sami I. Alzarea, Abdullah F. Alasmari.

**Formal analysis:** Abdulrahman Alshammari, Mashal M. Almutairi.

**Funding acquisition:** Abdullah F. Alasmari.

**Investigation:** Metab Alharbi, Mohamed A. Mahmoud, Abdulrahman Alshammari, Mashal M. Almutairi, Jihan M. Al-Ghamdi, Jawza F. Alsabhan, Othman Al Shabanah, Norah A. Alshalawi, Sami I. Alzarea.

**Methodology:** Metab Alharbi, Mohamed A. Mahmoud, Abdulrahman Alshammari, Mashal M. Almutairi, Jihan M. Al-Ghamdi, Jawza F. Alsabhan, Othman Al Shabanah, Norah A. Alshalawi, Sami I. Alzarea, Abdullah F. Alasmari.

**Resources:** Mohamed A. Mahmoud, Abdullah F. Alasmari.

**Software:** Abdulrahman Alshammari.

**Supervision:** Metab Alharbi.

**Validation:** Abdulrahman Alshammari, Mashal M. Almutairi, Jihan M. Al-Ghamdi.

**Writing – original draft:** Metab Alharbi, Mohamed A. Mahmoud, Abdulrahman Alshammari.

**Writing – review & editing:** Metab Alharbi, Mohamed A. Mahmoud, Abdulrahman Alsham-mari, Mashal M. Almutairi, Jihan M. Al-Ghamdi, Jawza F. Alsabhan, Othman Al Shabanah, Norah A. Alshalawi, Sami I. Alzarea, Abdullah F. Alasmari.

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
