## [Decision Letter · Decision Letter 0]

8 Sep 2024

PONE-D-24-32686The Ameliorative Effect of Pioglitazone Against Colistin-induced Nephrotoxicity is Mediated by Inhibition of NF-kB and Restoration of Nrf2 Signaling: An Integrative Bioinformatics Prediction-Guided In Vitro StudyPLOS ONE

Dear Dr. Alharbi,

Thank you for submitting your manuscript to PLOS ONE. After careful consideration, we feel that it has merit but does not fully meet PLOS ONE’s publication criteria as it currently stands. Therefore, we invite you to submit a revised version of the manuscript that addresses the points raised during the review process.

We look forward to receiving your revised manuscript.

Kind regards,

Roland Eghoghosoa Akhigbe

Academic Editor

PLOS ONE

Journal Requirements:

"The authors extend their appreciation to the Deputyship for Research & Innovation, “Ministry of Ed-ucation” in Saudi Arabia for funding this research work through the project number (IFK-SUDR_H162)"    

Reviewers' comments:

Reviewer's Responses to Questions

**Comments to the Author**

1. Is the manuscript technically sound, and do the data support the conclusions?

Reviewer #1: Yes

Reviewer #2: Yes

2. Has the statistical analysis been performed appropriately and rigorously? 

Reviewer #1: Yes

Reviewer #2: Yes

3. Have the authors made all data underlying the findings in their manuscript fully available?

Reviewer #1: Yes

Reviewer #2: Yes

4. Is the manuscript presented in an intelligible fashion and written in standard English?

Reviewer #1: Yes

Reviewer #2: Yes

5. Review Comments to the Author

Reviewer #1: Peer Review Report for Manuscript PONE-D-24-32686

Summary of the Research and Overall Impression

The manuscript titled "The Ameliorative Effect of Pioglitazone Against Colistin-induced Nephrotoxicity is Mediated by Inhibition of NF-κB and Restoration of Nrf2 Signaling: An Integrative Bioinformatics Prediction-Guided In Vitro Study" investigates the protective effects of pioglitazone against colistin-induced nephrotoxicity. The study employs a combination of bioinformatics predictions and in vitro experiments using HK-2 human kidney cells to explore the underlying mechanisms. The authors demonstrate that pioglitazone modulates NF-κB-mediated inflammatory signaling and Nrf2-mediated antioxidative stress pathways, leading to increased cell viability and reduced lactate dehydrogenase release in colistin-treated cells.

Overall, this study provides valuable insights into the potential therapeutic application of pioglitazone in preventing colistin-induced kidney damage. The integrative approach combining bioinformatics and experimental validation is a significant strength of this manuscript. However, several methodological and interpretative issues need to be addressed to strengthen the validity and impact of the findings.

Recommendation: Major Revision

Major Issues

1. Lack of Controls in Bioinformatics Predictions:

o The manuscript does not include appropriate controls in the bioinformatics analysis to ensure specificity. Including negative controls, such as unrelated drugs or conditions, would strengthen the claim that the observed effects are specific to the pioglitazone and colistin interaction (Introduction, Methodology).

2. Sample Size Justification:

o The manuscript lacks a clear justification for the sample sizes used in the in vitro experiments. A power analysis or rationale for the chosen sample sizes should be provided to ensure the statistical reliability of the findings (Methodology, Statistical Analysis).

3. Potential Overinterpretation of Findings:

o The manuscript suggests that pioglitazone restores Nrf2 signaling solely based on in vitro data. While promising, these findings should be presented more cautiously, acknowledging the need for further validation in in vivo models (Discussion).

4. Lack of Ethical Approval Statement:

o Although the study uses human cell lines, there is no mention of ethical approval for these experiments. A statement regarding ethical approval or exemption should be included to clarify compliance with ethical standards (Ethical Considerations).

Minor Issues

1. Redundancy and Conciseness:

o The manuscript contains redundant phrases, such as repeatedly stating "the ameliorative effect of pioglitazone against colistin-induced nephrotoxicity." The authors should streamline the text to improve readability (Introduction, Discussion).

2. Comprehensiveness of Literature Review:

o The literature review could be expanded to include more recent studies on colistin-induced nephrotoxicity and Nrf2 signaling in renal protection. This would provide a more comprehensive context for the study's findings (Introduction).

3. Availability of Raw Data:

o The manuscript does not mention the availability of raw data or supplementary materials. The authors should consider providing access to raw data and supplementary materials to enhance reproducibility (Methodology, Results).

Any Other Points

Confidential Comments to the Editors: The study's overall approach is sound, but the manuscript requires revisions to address improved clarity. There are no ethical concerns related to this study.

Reviewer #2: Dear Authors,

Congratulations on your significant findings. The work holds considerable value as it provides potential insights into effectively managing colistin-induced toxicity. However, to enhance the clarity and depth of your study, several nuanced areas require further attention:

1. Beyond nephrotoxicity, the NF-κB and Nrf2 pathways are involved in various cellular processes. Did your study demonstrate that the reported anti-inflammatory and antioxidative effects of Pioglitazone are specific to colistin-induced toxicity, or could these effects be general responses to other nephrotoxic agents? Were appropriate controls used to distinguish between specific and non-specific protective effects? To strengthen the study and confirm specificity, employing an inhibitor is recommended.

2. Given the interconnected nature of these pathways, Pioglitazone may directly or indirectly modulate them. Are there additional mechanisms, aside from those reported, that could contribute to Pioglitazone's nephroprotective effects? Were these potential mechanisms considered in your study?

3. Considering the implications of long-term Pioglitazone use, could chronic administration introduce complications not evident within the study's timeframe?

4. Additionally, with the growing concern over colistin resistance, Pioglitazone's nephroprotective effects might inadvertently encourage the continued use of colistin, potentially exacerbating antibiotic resistance issues. Could you discuss the broader implications of promoting Pioglitazone in this context?

5. Lastly, when extrapolating your findings, it is important to consider that Pioglitazone’s activation of PPARγ may have off-target effects, such as fluid retention or cardiovascular complications, which could counterbalance its nephroprotective benefits. Does your study account for these potential off-target effects? Addressing this is crucial, as any detrimental effects could outweigh the overall benefits of the drug.

These points should be addressed to improve the study's overall clarity and robustness.

6. PLOS authors have the option to publish the peer review history of their article (what does this mean?). If published, this will include your full peer review and any attached files.

Reviewer #1: **Yes: **Akorede Bolaji Adetibigbe

Reviewer #2: **Yes: **Ashonibare Victory J.

---

## [Author Response · Author response to Decision Letter 0]

22 Oct 2024

We have attached the responses to all reviewers in one file.

---

## [Decision Letter · Decision Letter 1]

6 Nov 2024

The Ameliorative Effect of Pioglitazone Against Colistin-induced Nephrotoxicity is Mediated by Inhibition of NF-kB and Restoration of Nrf2 Signaling: An Integrative Bioinformatics Prediction-Guided In Vitro Study

PONE-D-24-32686R1

Dear Dr. Alharbi,

We’re pleased to inform you that your manuscript has been judged scientifically suitable for publication and will be formally accepted for publication once it meets all outstanding technical requirements.

Kind regards,

Roland Eghoghosoa Akhigbe

Academic Editor

PLOS ONE

Reviewers' comments:

Reviewer's Responses to Questions

**Comments to the Author**

1. If the authors have adequately addressed your comments raised in a previous round of review and you feel that this manuscript is now acceptable for publication, you may indicate that here to bypass the “Comments to the Author” section, enter your conflict of interest statement in the “Confidential to Editor” section, and submit your "Accept" recommendation.

Reviewer #1: All comments have been addressed

Reviewer #2: All comments have been addressed

2. Is the manuscript technically sound, and do the data support the conclusions?

Reviewer #1: Yes

Reviewer #2: Yes

3. Has the statistical analysis been performed appropriately and rigorously? 

Reviewer #1: Yes

Reviewer #2: Yes

4. Have the authors made all data underlying the findings in their manuscript fully available?

Reviewer #1: Yes

Reviewer #2: Yes

5. Is the manuscript presented in an intelligible fashion and written in standard English?

Reviewer #1: Yes

Reviewer #2: Yes

6. Review Comments to the Author

Reviewer #1: (No Response)

Reviewer #2: (No Response)

7. PLOS authors have the option to publish the peer review history of their article (what does this mean?). If published, this will include your full peer review and any attached files.

Reviewer #1: **Yes: **Bolaji Akorede

Reviewer #2: **Yes: **Victory Ashonibare

---

## [Editor Report · Acceptance letter]

18 Nov 2024

PONE-D-24-32686R1 

PLOS ONE

Dear Dr. Alharbi, 

I'm pleased to inform you that your manuscript has been deemed suitable for publication in PLOS ONE. Congratulations! Your manuscript is now being handed over to our production team.

Kind regards, 

on behalf of

Dr. Roland Eghoghosoa Akhigbe 

Academic Editor

PLOS ONE